# Characteristics of COVID-19 Patients Admitted to Intensive Care Unit in Multispecialty Hospital of Riyadh, Saudi Arabia: A Retrospective Study

**DOI:** 10.3390/healthcare11182500

**Published:** 2023-09-08

**Authors:** Mansour Almuqbil, Ali Ibrahim Almoteer, Alwaleed Mohammed Suwayyid, Abdulaziz Hussain Bakarman, Raed Fawaz Alrashed, Majed Alrobish, Fahad Alasalb, Abdulaziz Abdulrahman Alhusaynan, Mohammed Hadi Alnefaie, Abdullah Saud Altayar, Saad Ebrahim Alobid, Moneer E. Almadani, Ahmed Alshehri, Adel Alghamdi, Syed Mohammed Basheeruddin Asdaq

**Affiliations:** 1Department of Clinical Pharmacy, College of Pharmacy, King Saud University, Riyadh 11451, Saudi Arabia; 2Department of Pharmacy Practice, College of Pharmacy, AlMaarefa University, Dariyah, Riyadh 13713, Saudi Arabia; aalmotair@ksu.edu.sa (A.I.A.); dr.alwaleedswayyid@gmail.com (A.M.S.); abdulazizh148@gmail.com (A.H.B.); raed.fa95@gmail.com (R.F.A.); magedarrow@hotmail.com (M.A.); k-s-a-909@hotmail.com (F.A.); phd.az95@gmail.com (A.A.A.); bn.hadi@gmail.com (M.H.A.); oooatooo@hotmail.com (A.S.A.); 3Department of Pharmacology and Toxicology, College of Pharmacy, King Saud University, Riyadh 11451, Saudi Arabia; saaalobaid@ksu.edu.sa; 4Department of Clinical Medicine, College of Medicine, AlMaarefa University, Dariyah, Riyadh 13713, Saudi Arabia; mmadani@mcst.edu.sa; 5Department of Pharmacology, College of Clinical Pharmacy, Imam Abdulrahman Bin Faisal University, King Faisal Road, Dammam 31441, Saudi Arabia; adalshehri@iau.edu.sa; 6Department of Pharmaceutical Chemistry, Faculty of Clinical Pharmacy, Al Baha University, P.O. Box 1988, Al Baha 65779, Saudi Arabia; ai.alghamdi@bu.edu.sa

**Keywords:** COVID-19, intensive care unit, Saudi Arabia, mortality, intubation, healthcare

## Abstract

During the early stages of the COVID-19 pandemic, infection rates were high and symptoms were severe. Medical resources, including healthcare experts and hospital facilities, were put to the test to ensure their readiness to deal with this unique event. An intensive care unit (ICU) is expected to be required by many hospitalized patients. Many hospitals worldwide lacked resources during the pandemic’s peak stages, particularly in critical care treatment. Because of this, there were issues with capacity, as well as an excessive influx of patients. Additionally, even though the research location provides medical care to a sizable population, there is a paucity of scientific data detailing the situation as it pertains to COVID-19 patients during the height of the outbreak. Therefore, this study aimed to identify and describe the features of COVID-19 patients hospitalized in the ICU of one of the multispecialty hospitals in Riyadh, Saudi Arabia. An observational retrospective study was conducted using a chart review of COVID-19 patients admitted to the ICU between March 2020 and December 2020. To characterize the patients, descriptive statistics were utilized. An exploratory multivariate regression analysis was carried out on the study cohort to investigate the factors that were shown to be predictors of death and intubation. Only 333 (29.33%) of the 1135 samples from the hospital’s medical records were used for the final analysis and interpretation. More than 76% of the patients in the study were male, with a mean BMI of 22.07 and an average age of around 49 years. The most frequent chronic condition found among the patients who participated in the study was diabetes (39.34%), followed by hypertension (31.53%). At the time of admission, 63 of the total 333 patients needed to have intubation performed. In total, 22 of the 333 patients died while undergoing therapy. People with both diabetes and hypertension had a 7.85-fold higher risk of death, whereas those with only diabetes or hypertension had a 5.43-fold and 4.21-fold higher risk of death, respectively. At admission, intubation was necessary for many male patients (49 out of 63). Most intubated patients had hypertension, diabetes, or both conditions. Only 13 of the 63 patients who had been intubated died, with the vast majority being extubated. Diabetes and hypertension were significant contributors to the severity of illness experienced by COVID-19 participants. The presence of multiple comorbidities had the highest risk for intubation and mortality among ICU-admitted patients. Although more intubated patients died, the fatality rate was lower than in other countries due to enhanced healthcare management at the ICU of the study center. However, large-scale trials are needed to determine how effective various strategies were in preventing ICU admission, intubation, and death rates.

## 1. Introduction

Coronavirus Disease 2019 (COVID-19) is an infectious disease that is caused by unique SARS-CoV-2 coronavirus strains. This disease is highly contagious [1]. It is a member of the large coronavirus family, commonly found in animals and humans. The virus can spread quickly from one species to another. Minor symptoms, such as rhinorrhea, headache, fever, coughing, and sore throat, may be the initial manifestations in immunocompetent persons. Still, they may eventually advance to more serious respiratory illnesses, such as acute bronchitis or pneumonia [2]. The most typical symptoms of this viral illness include fever, persistent cough, and shortness of breath. The incubation period for the disease is typically five days, and it can be active for more than eight days [3,4]. Men are more likely to be affected than women by this ailment, which has a mean age of 52, and one-third of patients have concomitant conditions, such as diabetes, cardiovascular disease, and hypertension [5].

It is estimated that twenty percent of hospitalized patients require the services of an intensive care unit (ICU), and the case fatality rate (CFR) is more than thirteen percent [4]. The infection’s documented frequent consequences include acute respiratory distress syndrome [6,7], acute cardiac injury [8,9], acute kidney injury [10,11], disseminated intravascular coagulation [12], and sepsis [13]. It has been found that patients who are older, male, and with comorbid diseases have higher rates of death while being treated in a hospital setting [14]. Many research studies conducted in a variety of nations concluded that certain independent risk factors, such as age and male sex, are linked to increased rates of death [15]. However, the primary risk factors for intubation and death differed among critically ill patients in those studies. These factors included hypertension [16], heart failure, a body mass index (BMI) greater than 40, diabetes [17,18,19], coronary artery disease, cerebrovascular disease [20], smoking, chronic obstructive pulmonary disease [21], and the simultaneous occurrence of three or more comorbidities [22]. There are studies [5,23] that document the association of age and gender with the severity of the COVID-19 illness, but there are a number of gray areas that need to be addressed to determine the involvement of factors such as race, ethnicity, the COVID-19 vaccine, specific underlying conditions, the use of specific medications to manage specific underlying diseases, laboratory test results, the type of COVID-19 regimen that is offered, and symptoms that are associated with severe outcomes among COVID-19 patients.

Saudi Arabia was quick to devise and put into action strategies to contain the spread of the COVID-19 pandemic. The Saudi Arabian government prepared public and private institutions to handle the pandemic with strategic preparation and a COVID-19 response plan, in accordance with the WHO operational planning recommendations to support the nation [24]. To track contacts, screen the population, raise awareness, and implement the necessary measures to stop the spread of this disease, the authorities established a governance framework made up of accountable committees. Treatment for COVID-19 was made available at no cost to citizens and residents of Saudi Arabia by the country’s health administration. The nation supported all COVID-19 medicine and vaccine proposals and kept all primary healthcare programs and immunization policies in place [25].

Due to the rapid increase in infections at the beginning of the pandemic, widespread shortages of personal protective equipment (PPE), diagnostic test kits, and crucial patient care equipment, such as respirators, were reported from many world regions [26]. Cross-contamination in molecular diagnostic laboratories and the lack of trained clinical personnel further posed challenges in COVID-19 management [27]. Further, many hospitals around the world suffered from a lack of resources, particularly in terms of the management of critical care. This resulted in problems with capacity and an excessive influx of patients. Hence, to maintain the intensive care unit (ICU) capacity, thorough triage is essential to identify patients who require immediate treatment. Although there have been studies in Saudi Arabia that show an association between chronic diseases and demographic characteristics of patients and the severity of COVID-19 infection [28,29], only a few studies have shown a link between chronic diseases, medication use, age, gender, and other demographic variables and intubation and mortality. Earlier, we showed that chronic morbidities in the Al-Ahsa region of Saudi Arabia increased the mechanical ventilation rate in COVID-19 patients [5]. There has not been any additional research from Saudi Arabia to ascertain whether there is a correlation between the patient’s characteristics and intubation and mortality. Therefore, being aware of these differences can aid in predicting which cases of COVID-19 will require intubation and can have higher chances of mortality, consequently improving the overall efficacy of case management [30,31,32,33]. This awareness also facilitates the development of preventative actions, the identification of COVID-19′s impact on hospital capacity, and the enhancement of patient bed capacity and healthcare systems through risk classification.

King Saud University Medical City is an academic medical center with experience in multi-facility and multi-disciplinary administration that provides tertiary treatment. It is on the King Saud University campus in Riyadh, Saudi Arabia’s biggest university. The world-class healthcare center employs more than 2072 allied health workers, 853 residents, and 1300 full-time physicians. Each year, they treat over a million outpatients and admit thousands. It served as a COVID-19 testing and vaccination facility. It received thousands of patients for COVID-19 management from March 2020 to December 2020. During this time, the country witnessed its highest peak of cases, and most residents and citizens were not immunized. Therefore, it is worthwhile to investigate the factors that may have influenced the COVID-19 management system. Hence, this study aimed to investigate the clinical and demographic characteristics of the COVID-19-infected patients admitted to the intensive care units, and their possible implications on the chances of intubation and mortality, through a retrospective study design.

## 2. Materials and Methods

### 2.1. Study Design and Settings

This was a chart review-based retrospective cohort study. Patients were included if they were admitted to King Saud University Medical City from March 2020 to December 2020, had a COVID-19 diagnosis (verified by patients diagnosed with SARS-CoV-2 by Real-Time Reverse Transcriptase Polymerase Chain Reaction), and were at least 18 years old. Patients with COVID-19 hospitalized in the intensive care unit were the only ones included. Patients who were not transferred to the intensive care unit or whose medical records were missing important details were not included in the study. The study was conducted in accordance with the Declaration of Helsinki and approved by the Institutional Review Boards of AlMaarefa University (IRB09-04122022-100, dated 4 December 2022) and King Saud University (IRB22/1057/IRB, dated 29 December 2022). The informed consent was waived due to the retrospective nature of the study.

### 2.2. Data Collection

Demographic characteristics (age, gender, and ethnicity) and other relevant factors (smoking status, height, and weight to calculate BMI) were acquired from electronic/non-electronic medical records and handled utilizing the hospital’s health information management (HIM) systems. The information on the status of comorbidities (diabetes, hypertension, established cardiovascular disease, ischemic heart disease, atrial fibrillation, end-stage renal disease, psychotic disorders, metabolic acidosis, deep vein thrombosis, chronic obstructive pulmonary disease, and others) was recorded. The status of medication use for managing the comorbidities (insulin, angiotensin-converting enzyme inhibitors, acetaminophen, non-steroidal anti-inflammatory drugs, loop diuretics, calcium channel blockers, aspirin, metformin, angiotensin receptor blockers, statin, thiazide diuretics, beta-blockers, albuterol, sitagliptin, amlodipine, and others) was recorded. Further, types of COVID-19 symptoms, such as fever, cough, difficulty breathing, headache, chills, myalgia, dizziness, nausea, diarrhea, and vomiting, were also obtained from the hospital medical record. Moreover, the details of the drugs (dexamethasone, azithromycin, hydroxychloroquine, methylprednisolone, tacilizumab, and others) used to treat COVID-19 were also acquired. Additionally, parameters such as ‘need for nasal cannula at admission’, ‘need for high-flow oxygen at the time of admission’, ‘need for intubation at admission and at a later stage’, radiological severity, lowest value of oxygen saturation, follow-up, and mortality status were also recorded. Following a review of the information gathered, only samples with data on each of the aforementioned factors were used in the study. Out of 1135 samples obtained from the medical records of the hospital, only 333 (29.33%) were used for final analysis and interpretation. The remaining samples were discarded due to incomplete information (Figure 1).

### 2.3. Data Analysis

The data obtained were entered into the statistical program developed by IBM called SPSS (version 25). Univariate descriptive analysis of the socio-demographic features of the study sample using Pearson Chi-square test was carried out. Multinomial regression analysis was performed to calculate the odds ratio as a measure of predictors for the dependent variables ‘intubation’ and ‘fatality’ in COVID-19 patients. To test the significance of the study’s findings for the appropriate degrees of freedom, the values were compared using a degree of significance of 0.05. The data were stratified to determine the influence of major characteristics of the patients on the outcome variables (intubation and mortality).

## 3. Results

### 3.1. Demographic Characteristics of the Patients

According to the data in Table 1, the average age of the participants was 49 years (ranging between 18 and 95 years with a standard deviation of 16.11, 48.74 ± 16.11). No significant difference was noticed between participants ≤ 50 years or older than 50. The number of patients with a BMI of more than 25 (overweight or obese) was not significantly different from the patients who had ≤25 (underweight or healthy weight) (55% vs. 45%). The average BMI was 22 (ranging between 18 and 28 with standard deviation of 8.04, 22.07 ± 8.04). On the contrary, a high percentage of the patients were of white ethnicity (90%), significantly different from the number of patients from other ethnicities. Further, a large proportion (77%) of patients included in the study were male (*p* = 0.001), and a significantly high number (89%) of the participants were non-smokers (*p* = 0.001).

### 3.2. Baseline Characteristics of the Patients

At the time of admission, 150 of the 333 patients in the study needed a nasal cannula. There was a significantly (*p* = 0.001) high number of patients who did not require high-flow oxygen at admission, with only 58 (17.42%) of the 333 patients requesting it. Furthermore, only 19% of the research participants were intubated, which is significantly (*p* = 0.001) smaller than the number of patients who were transferred to the critical care unit but not intubated. Furthermore, only 13.51 percent of the patients had radiological severity. A total of 2.77 percent of the patients were identified as having a very severe COVID-19 infection based on lung abnormalities (Table 2). Numerous study participants who were admitted to the ICU lacked any radiological severity.

### 3.3. COVID-19 Symptoms of The patients

Fever was the most reported symptom (72.67%) among the selected subjects, with cough and dyspnea coming in second and third place, respectively (Figure 2). Headache, chills, and myalgia were reported in 27.93%, 15.92%, and 1.41% of the recruited patients.

### 3.4. Status of Comorbidities of the Patients

Diabetes (39.34%) was the most common chronic condition among the people studied, followed by hypertension (31.53%). Bronchial asthma was the third most prevalent comorbidity observed (Appendix A). In addition, established cardiovascular illness, chronic renal disease, ischemic heart diseases, and hypothyroidism were the other comorbidities detected most frequently among the participants included in the study.

### 3.5. Description of Medications Used for Managing Comorbidities

Around one-quarter of the people who participated in the study utilized insulin to treat their comorbid conditions. Angiotensin-converting enzyme inhibitors (ACEI), acetaminophen, and prednisolone were the second, third, and fourth most taken drugs among the study participants, respectively (Appendix A). Additional drugs often found included calcium channel blockers (CCBs), loop diuretics, NSAIDs, and aspirin.

### 3.6. Description of Drugs Used for Managing COVID-19

Dexamethasone was administered to more than 60% of the patients to treat COVID-19 symptoms, whereas azithromycin was given to 36.34% (Appendix A). Moreover, hydroxychloroquine was prescribed to many patients, either alone (12.91%) or in combination with azithromycin (12.01%). Tocilizumab, prednisolone, and methylprednisolone were the other most frequently administered medicines in the study.

### 3.7. General Observations during Hospitalization and Follow-up

The average length of stay in the hospital was 13.43 days, with one patient staying for 182 days and another being discharged after one day. Only 55 patients, 17.68% of the total discharged patients, returned for follow-up. Due to the unique circumstances surrounding their comorbidities, only two of the 333 patients were required to remain in isolation, and another 63 of the 333 patients required intubation at the time of admission. Overall, 22 of the 333 patients passed away while receiving treatment.

### 3.8. Characteristics of Intubated Patients

Table 3 compares the major characteristics of intubated patients upon admission to the ICU with those of non-intubated ICU patients included in this study. The patients who needed to be intubated had an average age of 52 years and a standard deviation between 19 and 95 years old of 16.26. Although not statistically significant, more intubated patients (58.73%) than non-intubated patients (45.18%) were older than 50 years of age. Both groups had a similar proportion of male patients (76.77% vs. 76.29%). Fewer patients in both groups smoked cigarettes (7.93% vs. 12.22%). Compared to the group who were not intubated, a significantly higher percentage of diabetic, hypertensive, and patients with established cardiovascular disease were detected in the group who required intubation (*p* = 0.032). In contrast to the non-intubated group, a significant proportion of intubated patients had higher radiological severity (*p* = 0.021). Finally, the percentage of mortality among intubated patients was more significant (20.63 vs. 3.33%) than that among non-intubated patients (*p* = 0.049).

### 3.9. Characteristics of Deceased Patients

Table 4 demonstrates that, among the study samples, age had no significant (*p* = 078) effect on the mortality of patients admitted to the ICU. In our analysis, 91% of the deceased patients were male, significantly (*p* = 039) higher than the female patients. Smoking was significantly (*p* = 0.001) linked to patient mortality, with nearly 41% of the deceased patients smoking compared to only 9.37% of the patients who were discharged. Also, significantly (*p* = 0.022) more patients with diabetes, hypertension, asthma, and established cardiovascular diseases were found in the deceased group compared to the discharged group. Finally, compared to discharged samples, the radiological severity of the deceased patients was significantly higher (*p* = 0.011).

### 3.10. Predictors for Fatality in COVID-19 Patients

Comorbidities were the strongest predictors of mortality among the participants in our study. Individuals with both diabetes and hypertension had a 7.85-fold increased risk of mortality, whereas patients with only diabetes or hypertension had a 5.43-fold and 4.21-fold increased risk of death, respectively. Individuals with cardiovascular disease had an odds ratio of 3.81, and men had a 3.42-fold increased chance of dying from COVID-19 manifestations. The intubated patients showed a significantly increased mortality rate compared to patients who were not intubated (Table 5). Smoking increased mortality risk by 4.32 times compared to non-smokers in our study samples.

### 3.11. Predictors for Intubation in COVID-19 Patients

Diabetes was a significant predictor of intubation, either alone or in combination with hypertension, followed by hypertension and established cardiovascular diseases. In addition, individuals on insulin and furosemide had a significantly increased probability of requiring intubation (Table 6). In addition, male gender and advanced age of over 50 years were further predictors of intubation in the samples used for the study.

## 4. Discussion

This study aimed to assess the features of COVID-19 patients admitted to the ICU in Riyadh Hospital in 2020. Vaccines were not widely available during this period. As a result, the number of patients admitted to the ICU was relatively large. On 17 December 2020, vaccinations were first administered in Saudi Arabia [34]. Furthermore, the Riyadh region had the highest number of COVID-19-infected individuals [35], and multispecialty hospitals like our study center saw a high volume of patients during the peak time. Patients who were admitted to the hospital’s intensive care unit were included in the study. The study identified an association between comorbidities and the severity of COVID-19 symptoms and patient outcomes.

This study’s findings indicate that males have a 3.42 times higher risk of mortality than females. The number of males admitted to the ICU during the study period was significantly higher than that of females. Therefore, more patients taken for intubation were male than female. The findings of this study are consistent with those of earlier investigations. A single-arm meta-analysis [36] exhibited that men represent a significantly higher percentage of COVID-19 patients at 60%, and another study reported [37] a significantly higher number of men having the infection than women. In addition, earlier research has found that more men than women are infected with other coronaviruses, such as SARS-CoV [38] and MERS-CoV [39]. The findings of our study follow the same pattern as those seen in past investigations and are, therefore, not surprising. In most cases, females have a higher natural resistance to illnesses than their male counterparts. This may be caused by a combination of variables, including sex hormones, a high expression of coronavirus receptors (ACE 2) in men, and men’s lifestyle habits, which include greater rates of smoking and drinking than women. In addition, women generally have a more responsible attitude than males concerning the COVID-19 pandemic. This may reversibly impact implementing preventive measures, such as frequent hand washing, face masks, and stay-at-home orders [40].

There are conflicting data on the impact of ACE inhibitors and ARBs in addressing COVID-19 infection. Some reports indicate that ACEI and ARB play a protective role in preventing the severity of COVID-19 infection [41]. On the other hand, other reports indicate a significantly increased risk of severe or critical COVID-19 disease and ICU admission in patients receiving ACEI/ARB [42]. However, we could not identify an association between ACEI/ARB use and the severity of COVID-19, ICU hospitalization, intubation, or mortality in our study participants.

Our research also shows that patients who required mechanical ventilation were more likely to die than those who did not. However, the fatality rate in the same population is still significantly lower than reported in the relevant literature. Research in Mexico [43] found that mechanically ventilated patients had a mortality rate of 73.7%, whereas another study in Spain [44] found that mechanical ventilation improved survival. The difference in fatality rate could be attributable to various factors, including the quality of treatment provided, changes in the environment and geography, and the presence of a concurrent illness. In Saudi Arabia, The National Strategy to Standardize and Enhance Mechanical Ventilation project was launched in 2019 to improve patient care through evidence-based procedures [45]. There is the potential for higher standards of medical treatment to be practiced in the Kingdom. In addition, recent attempts made by the authorities to gain certification in terms of accreditation for their standard procedures have also contributed to improving the quality of the services. The Central Board of Accreditation for Healthcare Institutions has accredited 65 Ministry of Health (Saudi Arabia) hospitals [46]. In addition, several private and public hospitals have earned international accreditations, such as that of the Joint Commission International, which the International Society grants for Accreditation of Quality in Health Care, which the World Health Organization designates as a safety collaborating center [6]. Joint Commission International has granted accreditation to around 75 different hospitals in Saudi Arabia [47]. Accreditation of hospitals has a beneficial effect on most patient safety indicators. It is, therefore, one of the driving forces toward enhancing healthcare quality in Saudi Arabia [48].

The mean age of the participants was 48 years, with a max of 95 years and a minimum of 18 years. The median age of the patients included was 51 years. Our study samples were both advanced-aged and critically ill; around 45% needed a nasal cannula, and 19% needed intubation at admission, indicating their critical condition. Radiologically, the severity of illness was found in around 16% of the participants. The study by Khan et al. [49] from Saudi Arabia found that mechanically ventilated patients have 5 times the risk of death compared to non-mechanically ventilated patients.

The impact of chronic diseases on COVID-19 prognosis is well established [50]. Conditions such as heart disease, diabetes, cancer, chronic renal disease, and obesity raise the risk of severe COVID-19-related illness [51]. In agreement with the earlier findings and studies, common comorbid conditions in our cohort were diabetes (39.34%), hypertension (31.53%), asthma (13.51%), cardiac disease (3.6%), and chronic kidney disease (3.6%). This indicates that conditions such as diabetes and hypertension are common risk factors in COVID-19 patients for ICU transfer. Further, more than 57% of the patients who needed intubation at admission were diabetic, and almost 50% were hypertensive. Also, it is important to know that 54.54% of the deceased COVID-19 patients had diabetes and hypertension together, while 36% had established cardiovascular diseases. This indicated a strong association between critical illness in COVID-19 patients and diabetes and hypertension.

Another intriguing finding of this study was the link between the likelihood of intubation and certain drugs (insulin and furosemide) used to treat chronic conditions. The findings of past studies are consistent with this one. According to the findings of Yang et al. [52]’s meta-analysis, insulin usage is related to a mortality risk of approximately two times higher (odds ratio = 2.10; 95% CI: 1.51–2.93). Another fascinating discovery was the role of loop diuretics as a predictor of intubation. Furosemide may alter dyspnea mechanistically, in addition to its effects as a diuretic, due to its effect on the release of pro-inflammatory cytokines (interleukin-9, interleukin-8, and tumor necrosis factor) [53]. This action is independent of the diuretic effects of furosemide. The effectiveness of furosemide as a possible treatment for COVID-19 is now being investigated [54].

Although the study’s main goals were attained, some limitations must be addressed before proceeding with the current findings. Data from 2020 are presented in this analysis; at that time, a COVID-19 wild-type variant was prevalent and was held accountable for the rapid spread of the illness and poor clinical outcomes [29]. Because different COVID-19 variants differed in terms of transmissibility and pathogenicity, our study results would have been different if it had been undertaken during the period of omicron dominance. For that reason, the results of this study should be viewed with caution, as they cannot be generalized to other time periods. Further, we were only able to rely on the records available at the hospital. Therefore, there is a possibility of bias occurring as a result of the retrospective aspect of the study. Additionally, because just one hospital is included in the study sample, the investigation’s single-centric design prevents results from being generalized to the entire population. Because some information was missing, only 333 of the 1135 files could be used for the analysis. Since certain vital pieces of information were missing from those files, and it was impossible to avoid this prejudice. This has caused a reduction in the sample size of this study, which is another limitation. Furthermore, because vaccination was unavailable [55] during the study period, the impact of vaccination status cannot be investigated. As evidenced by multiple studies [56,57], vaccinations have significantly decreased the severity and manifestation of COVID-19 patients, particularly in susceptible populations; therefore, it is critical to develop effective vaccination strategies for controlling infectious diseases such as COVID-19. We anticipate that the COVID-19-infected patients hospitalized after vaccination will experience noticeably reduced severe intubation and death rates. Therefore, it is recommended to carry out a comprehensive study of the characteristics of COVID-19 patients when the vaccination was available in the Kingdom of Saudi Arabia. In addition, because the study was based solely on medical records from 2020, it cannot accurately reflect the current state of the medical facility.

## 5. Conclusions

Diabetes and hypertension were major factors in increasing the severity of illness in COVID-19 patients in our study samples. Smoking and male gender were other major predictors of higher mortality rates in the study population. Although more intubated patients died, the fatality rate was still lower than in other countries due to the advanced ICU healthcare management and good hospital preparation. Further, large-scale studies are required to understand how protective other interventions were in preventing ICU admission, intubation, and fatality rate. Also, it would be interesting to know the characteristics of the COVID-19 patients post-vaccination.

## Figures and Tables

**Figure 1 healthcare-11-02500-f001:**
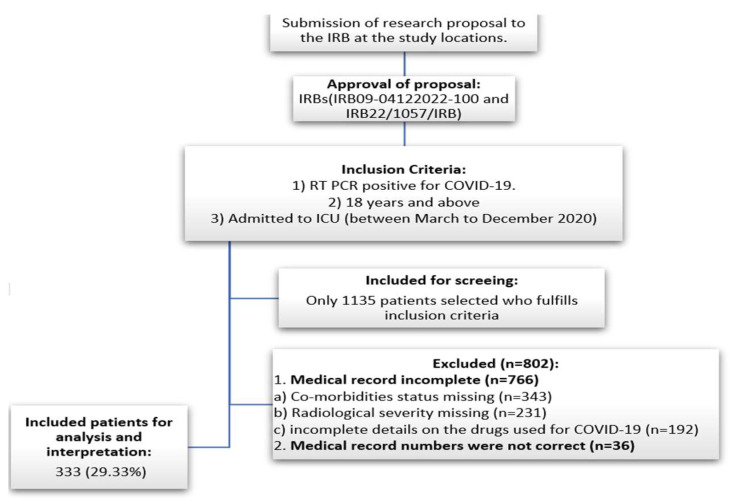
Patient selection and work flowchart.

**Figure 2 healthcare-11-02500-f002:**
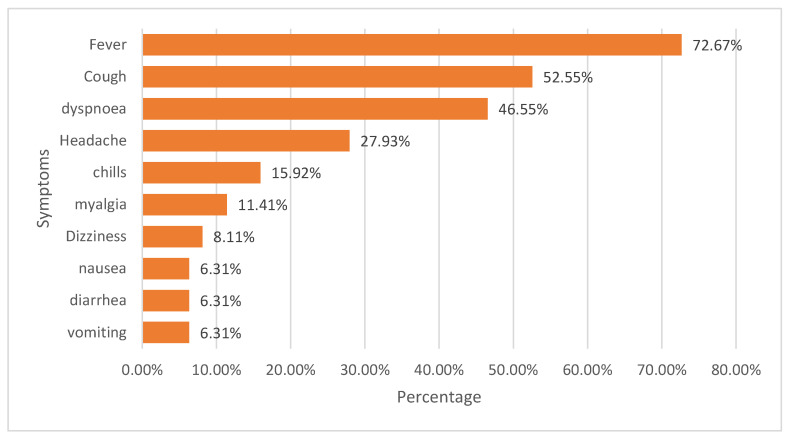
Percentage distribution of symptoms.

**Table 1 healthcare-11-02500-t001:** Demographic characteristics of the patients (n = 333).

Characteristics	Variables	Number (%)	*p* Value *
Age (years)	≤50	173 (52)	0.645
	>50	160 (48)	
BMI	≤25	151 (45)	0.382
	>25	182 (55)	
Ethnicity	White	299 (90)	0.001
	Black	30 (9)	
	Others	04 (1)	
Gender	Male	255 (77)	0.001
	Female	78 (23)	
Smoking	Yes	38 (11)	0.001
	No	295 (89)	

* Chi-square test or Fischer test, whichever is applicable.

**Table 2 healthcare-11-02500-t002:** Baseline general characteristics of the patients.

Characteristics	Variables	Frequency	Percentage	*p* Value *
Need for the nasal cannula	Yes	150	45.05%	0.341
	No	183	54.95%	
Need for high-flow oxygen on admission	Yes	58	17.42%	0.001
	No	275	82.58%	
Need for intubation on admission.	Yes	63	18.92%	0.001
	No	270	81.08%	
Radiological severity of patients	Mild	179	53.75%	0.041
	Moderate	100	30.03%	
	Severe	45	13.51%	
	Very Severe	9	2.70%	

* Chi-square test or Fischer test, whichever is applicable.

**Table 3 healthcare-11-02500-t003:** Comparison of the characteristics of the patients intubated and non-intubated.

Characteristics	Intubated (63), n (%)	Non-Intubated (270), n (%)	*p* Value *
Age (years)			0.067
≥50	37 (58.73)	122 (45.18)	
˂50	26 (41.26)	148 (54.81)	
Gender			0.156
Male	49 (77.77)	206 (76.29)	
Female	14 (22.22)	64 (23.70)	
Smoking status			0.132
Yes	5 (7.93)	33 (12.22)	
No	58 (92.06)	237 (87.77)	
Chronic disease status			0.032
Diabetes	36 (57.14)	96 (35.55)	
Hypertension	31 (49.20)	74 (27.40)	
Asthma	08 (12.69)	37 (13.70)	
Established cardiovascular diseases	08 (12.69)	06 (2.22)	
Radiological severity			0.021
Mild	10 (15.87)	169 (62.59)	
Moderate	21 (33.33)	79 (29.25)	
Severe	24 (38.09)	21 (7.77)	
Very severe	08 (12.69)	01 (.37)	
Mortality status			0.049
Yes	13 (20.63)	09 (3.33)	
No	50 (79.36)	261 (96.66)	

* Chi-square test or Fischer test, whichever is applicable.

**Table 4 healthcare-11-02500-t004:** Comparison of the characteristics of the deceased and discharged patients.

Characteristics	Deceased (22), n (%)	Discharged (311), n (%)	*p* Value *
Age (years)			0.078
≥50	13 (59.09)	146 (46.94)	
˂50	09 (40.90)	165 (53.05)	
Gender			0.039
Male	20 (90.90)	235 (75.56)	
Female	02 (9.10)	76 (24.43)	
Smoking status			0.001
Yes	09 (40.90)	29 (9.37)	
No	13 (59.09)	282 (90.67)	
Chronic disease status			0.022
Diabetes	12 (54.54)	119(38.26)	
Hypertension	12 (54.54)	94 (30.22)	
Asthma	04 (18.18)	41 (13.18)	
Established cardiovascular diseases	08 (36.36)	06 (1.92)	
Radiological severity			0.011
Mild	04 (18.18)	175 (56.27)	
Moderate	02 (9.10)	98 (31.51)	
Severe	10 (45.45)	35 (11.25)	
Very severe	06 (27.27)	03 (.96)	

* Chi-square test or Fischer test, whichever is applicable.

**Table 5 healthcare-11-02500-t005:** Predictors for fatality in COVID-19 ICU admitted patients.

Characteristics	Odds Ratio	Confidence Interval		*p* Value
		Lower	Upper	
Sex	3.42	2.38	5.49	0.021
Intubation	1.67	1.21	2.34	0.034
Diabetes and hypertension	7.85	4.31	9.32	0.001
Diabetes	5.43	3.21	7.21	0.001
Hypertension	4.21	2.98	5.32	0.001
Established CV diseases	3.81	2.87	5.98	0.001
Smoking	4.32	2.11	6.21	0.001

**Table 6 healthcare-11-02500-t006:** Predictors for intubation in COVID-19 ICU admitted patients.

Characteristics	Odds Ratio	Confidence Interval		*p* Value
		Lower	Upper	
Sex	2.92	1.88	3.69	0.021
Diabetes and hypertension	6.29	4.98	8.22	0.001
Diabetes	5.64	3.66	6.55	0.001
Hypertension	4.71	2.83	5.72	0.001
Established CV diseases	1.91	1.07	2.76	0.001
Insulin	3.93	2.79	5.77	0.001
Furosemide	3.65	2.81	4.89	0.001
Age more than 50 years	1.81	1.76	3.21	0.034

## Data Availability

Data is available with the corresponding author.

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
