# Peer review of "Characteristics of COVID-19 Patients Admitted to Intensive Care Unit in Multispecialty Hospital of Riyadh, Saudi Arabia: A Retrospective Study"

_healthcare, 2023, doi:10.3390/healthcare11182500_

Round 1

Reviewer 1 Report

The study in question is retrospective, monocentric (low degree of scientific evidence) 

with a huge selection bias (only 333/1135 patients were analyzed). Recently, many articles have been published about covid19- age-gender-intubation-mortality and the results do not carry anything new to the numerous world literature. 

good

Author Response

Response to the comments of Reviewer 1

Comments and Suggestions for Authors

The study in question is retrospective, monocentric (low degree of scientific evidence) 

with a huge selection bias (only 333/1135 patients were analyzed). Recently, many articles have been published about covid19- age-gender-intubation-mortality and the results do not carry anything new to the numerous world literature. 

Response: We value the respected reviewer's viewpoint. Due to the fact that this study relied on data from the medical record during the peak of COVID 19 cases, and hospital resources were scarce, the manual entry of all essential information was not available in the record resulted in missing some information as per our study requirement, leading to in the filtration of 802 samples. However, the samples available were sufficient to assess the study's findings.

We acknowledge that there are already numerous studies on COVID-19- age-gender-intubation-mortality; however, there is a dearth of articles focusing on this topic from this specific location (Saudi Arabia), and the addition of data from the new sample sites will only serve to enrich the existing body of research. The study's findings therefore have a novel element.

Comments on the Quality of English Language

good

Reviewer 2 Report

The manuscript by Almuqbil et al. analyzes the characteristics of COVID-19 Patients Admitted to Intensive Care Unit in the multispecialty Hospital of Riyadh, Saudi Arabia. The authors showed that the presence of multiple co-morbidities had the highest risk for intubation and mortality among ICU-admitted patients. Although more intubated patients died, the fatality rate was lower than in other countries due to enhanced healthcare management at the ICU of the study center. The article is interesting and sound. However, I have some issues to report:

- Please remove all the abbreviations that do not appear more than three times.

- Line 69-71. Authors should provide adequate references for this statement. In fact, several studies have been conducted on the effect of COVID-19 on lungs (doi: 10.5694/mja2.50674 - doi: 10.1016/j.ccc.2021.05.003), cardiovascular system (doi: 10.1007/s00134-023-07147-z - doi: 10.1111/echo.15462), kidneys (doi: 10.1016/j.ccc.2022.01.002 - doi: 10.1111/aor.14078) and coagulation (doi: 10.1016/j.mayocp.2020.10.031). Please add these references to the list.

- Line 127-129. It is not clear whether those are exclusion criteria. Please explain.

- Line 131. Please provide the date of approval.

- Figure 2. Please replace difficulty breathing with "dyspnoea".

- Line 283- Meta-analysis is misspelled.

- The result section is hard to follow due to the high number of tables and figures, which seem sometimes unnecessary. Please delete table 4 and provide Figure 3, Figure 4 and Figure 5 and Table 7 as supplementary material.

- Table 8. What is level 1 and level 2? There is no need to specify as it is clear in the first column. Please delete the 2 columns of level 1 and 2.

- Please add the low number of patients as a limitation of the study.

- In the conclusion section, authors should stress out that those results were limited to their study population.

Author Response

Response to the comments of Reviewer 2

Comments and Suggestions for Authors

The manuscript by Almuqbil et al. analyzes the characteristics of COVID-19 Patients Admitted to Intensive Care Unit in the multispecialty Hospital of Riyadh, Saudi Arabia. The authors showed that the presence of multiple co-morbidities had the highest risk for intubation and mortality among ICU-admitted patients. Although more intubated patients died, the fatality rate was lower than in other countries due to enhanced healthcare management at the ICU of the study center. The article is interesting and sound. However, I have some issues to report:

- Please remove all the abbreviations that do not appear more than three times.

Response: Thank you very much respected reviewer, we have checked the whole manuscript and removed abbreviations that are not repeated more than three times.

- Line 69-71. Authors should provide adequate references for this statement. In fact, several studies have been conducted on the effect of COVID-19 on lungs (doi: 10.5694/mja2.50674 - doi: 10.1016/j.ccc.2021.05.003), cardiovascular system (doi: 10.1007/s00134-023-07147-z - doi: 10.1111/echo.15462), kidneys (doi: 10.1016/j.ccc.2022.01.002 - doi: 10.1111/aor.14078) and coagulation (doi: 10.1016/j.mayocp.2020.10.031). Please add these references to the list.

Response: Thanks for your recommendation. We have included all suggested references.

- Line 127-129. It is not clear whether those are exclusion criteria. Please explain.

Response: Yes, it is an exclusion criterion. The statement is now rephrased to improve clarity.

- Line 131. Please provide the date of approval.

Response: Included now

- Figure 2. Please replace difficulty breathing with "dyspnoea".

Response: Difficulty breathing has now changed to “dyspnoea” as per the reviewer’s recommendation.

- Line 283- Meta-analysis is misspelled.

Response: Corrected now

- The result section is hard to follow due to the high number of tables and figures, which seem sometimes unnecessary. Please delete table 4 and provide Figure 3, Figure 4 and Figure 5 and Table 7 as supplementary material.

Response: Thanks for your comments. As per the recommendation, Table 4 is now deleted, and other tables are re-numbered. Table 1 is inserted in the text (as per the comment of another reviewer).

In the current revised version, Table 5 explains the predictors for fatality, and Table 6 predictors for intubation. As these are outcome variables of the study, the authors believe that it would be good if we could retain them in the main document.

Further, Figures 3, 4, and 5 are now shifted to supplementary material as Supplementary Figure 1, Supplementary Figure 2, and Supplementary Figure 3, respectively.

- Table 8. What is level 1 and level 2? There is no need to specify as it is clear in the first column. Please delete the 2 columns of level 1 and 2.

Response: Thanks for your comment.

We have deleted the columns of levels 1 and 2 now in Table 7 (previously Table 8).

- Please add the low number of patients as a limitation of the study.

Response: Included now

- In the conclusion section, authors should stress out that those results were limited to their study population.

Response: Thanks for your advice. Included now.

Reviewer 3 Report

The manuscript healthcare-2566856 “Characteristics of COVID-19 Patients Admitted to Intensive Care Unit in Multispecialty Hospital of Riyadh, Saudi Arabia: 3 A retrospective study” proposes to identify and describe the features of COVID-19 patients hospitalized in the intensive care unit of one of the multispecialty hospitals in Riyadh, Saudi Arabia. This study described that the country experienced its largest peak of cases during this time, while most residents and citizens were unvaccinated. Therefore, it is worthwhile to investigate the factors that may have influenced the COVID-19 management system. Although the authors try to indicate the importance of this study, it seems to me that in the present format, the analysis seeks to meet a local demand, of greater interest to the Intensive Care Unit in Multispecialty Hospital of Riyadh. Considering that there are dozens of similar studies showing similar characteristics of the unicentric care of patients with COVID-19, it is necessary to point out some unique aspects of this hospital or Saudi Arabia’s measures against COVID-19, so that the justificative of the work is of interest to a greater number of readers. Adequately, the authors inform that there hasn't been any additional research from Saudi Arabia to ascertain whether there is a correlation between the patient's characteristics and intubation and mortality, I suggest that this motivation of this study should be improved and inserted in the abstract. Is it possible to compare the present data with other COVID-19 waves in the same Hospital?

The field present manuscript is appropriated for Healthcare (ISSN 2227-9032). A limitation of this study was the presence of only 333 (29.33%) of the 1135 samples from the hospital's medical records that were used for the final analysis and interpretation.

Following other studies in the literature, the present manuscript also indicates that COVID-19 patients with diabetes and hypertension had a ~8-fold higher risk of death, whereas those with only diabetes and hypertension had a higher risk of death. At admission, intubation was necessary for the majority of male patients (49 out of 63). Most intubated patients had hypertension, diabetes, or both conditions. Only 13 of the 63 patients who had been intubated died, with the vast majority being extubated. Diabetes and hypertension were significant contributors to the severity of illness experienced by COVID-19 participants. The presence of multiple co-morbidities had the highest risk for intubation and mortality among ICU-admitted patients.

The next comments should be addressed.

Major comments:

-          In the abstract section: Please, insert a clear motivation for this study. There are dozens of similar articles with a single-center demographic analysis of COVID-19. Clearly indicate the originality of this work.

-          In the introduction: While the motivation or uniqueness of this study must be strengthened, the introduction is adequate for the job by providing key information to readers. Please rewrite this sentence in L114 - "The country experienced its largest peak of cases during this time, while most 114 residents and citizens were unvaccinated".

-          In the material and methods section: In my point of view table 1 is incomplete or could be removed to insert text. No quantitative data is presented in Table 1. The word "number" in this table seems meaningless. About 70% of the samples were removed from the analysis and the inclusion criteria are quite comprehensive. It would be possible to explain the removal of such a large number of patients.

-          In the Discussion section: Although in the discussion, this study makes comparisons with other similar studies in the literature that were carried out in other countries, the study must point out unique characteristics of this study to increase the originality of the present study.

-          In the conclusion: The authors stated that “Although more intubated patients died, the fatality was still lower than in other countries due to the advanced ICU healthcare management and good hospital preparation.”. Considering that this comparison was not present in the results sections, this comparison should be stressed in the discussion section to be present in the conclusion.

It was described previously.

Author Response

Response to the comments of Reviewer 3

Comments and Suggestions for Authors

The manuscript healthcare-2566856 “Characteristics of COVID-19 Patients Admitted to Intensive Care Unit in Multispecialty Hospital of Riyadh, Saudi Arabia: 3 A retrospective study” proposes to identify and describe the features of COVID-19 patients hospitalized in the intensive care unit of one of the multispecialty hospitals in Riyadh, Saudi Arabia. This study described that the country experienced its largest peak of cases during this time, while most residents and citizens were unvaccinated. Therefore, it is worthwhile to investigate the factors that may have influenced the COVID-19 management system. Although the authors try to indicate the importance of this study, it seems to me that in the present format, the analysis seeks to meet a local demand, of greater interest to the Intensive Care Unit in Multispecialty Hospital of Riyadh. Considering that there are dozens of similar studies showing similar characteristics of the unicentric care of patients with COVID-19, it is necessary to point out some unique aspects of this hospital or Saudi Arabia’s measures against COVID-19, so that the justificative of the work is of interest to a greater number of readers. Adequately, the authors inform that there hasn't been any additional research from Saudi Arabia to ascertain whether there is a correlation between the patient's characteristics and intubation and mortality, I suggest that this motivation of this study should be improved and inserted in the abstract. Is it possible to compare the present data with other COVID-19 waves in the same Hospital?

Response: Thanks for your observation and comments. The description of the measures taken by Saudi Arabia is now included from Line 87 to 96.

The unique nature of the hospital is described in the last paragraph of the introduction, Line 119 to 127.

Now we have added the motivation of this study in the abstract.

The field present manuscript is appropriated for Healthcare (ISSN 2227-9032). A limitation of this study was the presence of only 333 (29.33%) of the 1135 samples from the hospital's medical records that were used for the final analysis and interpretation.

Following other studies in the literature, the present manuscript also indicates that COVID-19 patients with diabetes and hypertension had a ~8-fold higher risk of death, whereas those with only diabetes and hypertension had a higher risk of death. At admission, intubation was necessary for the majority of male patients (49 out of 63). Most intubated patients had hypertension, diabetes, or both conditions. Only 13 of the 63 patients who had been intubated died, with the vast majority being extubated. Diabetes and hypertension were significant contributors to the severity of illness experienced by COVID-19 participants. The presence of multiple co-morbidities had the highest risk for intubation and mortality among ICU-admitted patients.

The next comments should be addressed.

Major comments:

-          In the abstract section: Please, insert a clear motivation for this study. There are dozens of similar articles with a single-center demographic analysis of COVID-19. Clearly indicate the originality of this work.

Response: Thanks for your suggestion.

Although there were many studies done in different parts of the world, no scientific literature was available specifically from the study site that analyzes the characteristics of the patients at the peak time of the COVID-19 pandemic, especially for those who were admitted to ICU.

-          In the introduction: While the motivation or uniqueness of this study must be strengthened, the introduction is adequate for the job by providing key information to readers. Please rewrite this sentence in L114 - "The country experienced its largest peak of cases during this time, while most 114 residents and citizens were unvaccinated".

Response: Thanks for your comment. The sentence is corrected now. Digit 114 does not come in between the sentences.

-          In the material and methods section: In my point of view table 1 is incomplete or could be removed to insert text. No quantitative data is presented in Table 1. The word "number" in this table seems meaningless. About 70% of the samples were removed from the analysis and the inclusion criteria are quite comprehensive. It would be possible to explain the removal of such a large number of patients.

Response: Table 1 is now removed and inserted in the text as per the comment of the reviewer.

Due to the fact that this study relied on data from the medical record during the peak of COVID 19 cases, and hospital resources were scarce, the manual entry of all essential information was not available in the record resulted in missing some information as per our study requirement, leading to in the filtration of 802 samples. However, the samples available were sufficient to assess the study's findings.

-          In the Discussion section: Although in the discussion, this study makes comparisons with other similar studies in the literature that were carried out in other countries, the study must point out unique characteristics of this study to increase the originality of the present study.

Response: Thanks for your valuable comment.

The unique characteristics of this study are outlined from lines 334 to 354. The fatality rate among the intubated patients was significantly lower at the study site compared to earlier reports. This could be attributed to improved hospital facilities.

More details are available in the text…….

-          In the conclusion: The authors stated that “Although more intubated patients died, the fatality was still lower than in other countries due to the advanced ICU healthcare management and good hospital preparation.”. Considering that this comparison was not present in the results sections, this comparison should be stressed in the discussion section to be present in the conclusion.

Response: This comparison is available in the discussion section from lines 334 to 354. This is also discussed in section 3.8, Table 3 (Results section).

Comments on the Quality of English Language

It was described previously.

Reviewer 4 Report

Dear Authors,

I was quite interested in this article. It discusses the topic of COVID-19 in ICUs, concentrating on several key points.

General evaluations: the manuscript should be revised. Good the Introduction and Discussion sections, even though some references are required. Materials and Methods and Results sections are to be revised.

Specifically:

Introduction

Lines 79-83: This statement is not totally correct in my opinion. It is true that there are still unanswered questions, but it is also true that the link with gender and age, as well as ethnicity and COVID-19 vaccinations, is not totally unknown, not to mention the effect of comorbidities. It should be changed, and references should be included. I recommend a paper in which these associations are well established (https://doi.org/10.1007/s12016-022-08921-5). Perhaps you should modify this part.

Materials and Methods

Table 1. I'm not sure what the point of this table is, let alone the "Number" column, which is empty. I'd like you to have a look and assess its usefulness. Also, DVT and COPD should be in parentheses rather than with a hyphen.

Results

You become lost because there are so many sections. Furthermore, many of these are redundant. As many of these sections are descriptive, consider their appropriateness. Consider retaining only those that effectively add.

Furthermore, I was perplexed by how you handled the multivariate analyses (Table 7 and Table 8). What dependent variable did you investigate? What independent variables did you employ in your analysis? Why is sex listed as "Level 1" and "Level 2" but not as a "Characteristic"? Why was age not taken into account in Table 7? What methodology has been used? The tables are unclear, and I would like a complete response to these assessments.

Discussion

Line 271: "Generally"? What exactly do you mean? When did the first COVID-19 vaccines become available in Saudi Arabia?

Line 272: state that the first administrations began on October 17, 2020. First, I would like you to cite the source; I also wonder if the date is not too early. As far as I know, the first COVID-19 vaccine was administered on December 8, 2020, in the US.

Lines 328-330: Regarding these factors, although you have not highlighted them, I believe that comments on the clinical severity of the several SARS-CoV-2 variants should be made. After all, your assessments were done in 2020, when the Wild-Type and a few other variants were in circulation. Other strains with distinct transmissibility and pathogenicity features took over in the years that followed. This part, I believe, should be included in the debates, suggesting an article that makes these judgments based on population data (https://doi.org/10.3390/v15010125).

I also believe that the fact that we have focused on 2020 as a time frame should be considered a limitation, and that the considerations of what has been mentioned cannot be confirmed as a result.

Author Response

Response to the comments of Reviewer 4

Comments and Suggestions for Authors

Dear Authors,

I was quite interested in this article. It discusses the topic of COVID-19 in ICUs, concentrating on several key points.

General evaluations: the manuscript should be revised. Good the Introduction and Discussion sections, even though some references are required. Materials and Methods and Results sections are to be revised.

Response: Thanks for taking out your precious time and reviewing the manuscript. We have updated the manuscript now based on the comments of respected reviewers.

Specifically:

Introduction

Lines 79-83: This statement is not totally correct in my opinion. It is true that there are still unanswered questions, but it is also true that the link with gender and age, as well as ethnicity and COVID-19 vaccinations, is not totally unknown, not to mention the effect of comorbidities. It should be changed, and references should be included. I recommend a paper in which these associations are well established (https://doi.org/10.1007/s12016-022-08921-5). Perhaps you should modify this part.

Response: Thank you so much for your guidance. We have rewritten the statement and also included recommended citation.

Materials and Methods

Table 1. I'm not sure what the point of this table is, let alone the "Number" column, which is empty. I'd like you to have a look and assess its usefulness. Also, DVT and COPD should be in parentheses rather than with a hyphen.

Response: Thanks for your comment. We removed Table 1 and inserted the material in the text within section 2.2.

Results

You become lost because there are so many sections. Furthermore, many of these are redundant. As many of these sections are descriptive, consider their appropriateness. Consider retaining only those that effectively add.

Response: Thanks for your observation.

Tables and figures are re-arranged in this section. Figures 3, 4, and 5 have now shifted to supplementary materials.  Table 4 is deleted, and all other tables are re-numbered accordingly. Hope now the arrangement of tables and figures is clear to the respected reviewer.

Furthermore, I was perplexed by how you handled the multivariate analyses (Table 7 and Table 8). What dependent variable did you investigate? What independent variables did you employ in your analysis? Why is sex listed as "Level 1" and "Level 2" but not as a "Characteristic"? Why was age not taken into account in Table 7? What methodology has been used? The tables are unclear, and I would like a complete response to these assessments.

Response: Tables 7 and 8 are now numbered as Tables 5 and 6.  

Table 5 dependent variable was fatality as the title of the table states “Predictors for fatality in COVID-19 ICU admitted patients”.

Table 6 dependent variable was intubation as the title of the table states “Predictors for intubation in COVID-19 ICU admitted patients”.

All factors obtained from the medical record including age, gender, co-morbidity, medication use, smoking status, and other factors as mentioned in section 2.2 were included as independent factors. However, we have mentioned in Tables 5 and 6, only those factors that have shown significant effect on the outcome variable (dependent variable-fatality and intubation in Tables 5 and 6, respectively)

In Table 7 (now 5), age was not taken as it has no significant impact on the dependent variable (fatality), whereas age (more than 50 years) has shown a significant impact on intubation (Table 6).

We have removed Level 1 and level 2 columns from tables 5 and 6 as per the comment of another reviewer. The comment was the following:

- Table 8. What is level 1 and level 2? There is no need to specify as it is clear in the first column. Please delete the 2 columns of level 1 and 2.

Hope now we are successful in addressing the concerns of the respected reviewer about the analysis of Tables 5 and 6.

Discussion

Line 271: "Generally"? What exactly do you mean? When did the first COVID-19 vaccines become available in Saudi Arabia?

Response: We have corrected it now.

Vaccines were not widely available during this period.

On December 17, 2020, vaccinations were first administered in Saudi Arabia.

Line 272: state that the first administrations began on October 17, 2020. First, I would like you to cite the source; I also wonder if the date is not too early. As far as I know, the first COVID-19 vaccine was administered on December 8, 2020, in the US.

Response: Thanks for your observation. We regret that it was a typo error. It is supposed to be December 17, 2020.

Lines 328-330: Regarding these factors, although you have not highlighted them, I believe that comments on the clinical severity of the several SARS-CoV-2 variants should be made. After all, your assessments were done in 2020, when the Wild-Type and a few other variants were in circulation. Other strains with distinct transmissibility and pathogenicity features took over in the years that followed. This part, I believe, should be included in the debates, suggesting an article that makes these judgments based on population data (https://doi.org/10.3390/v15010125).

 Response: Thank you so much for this recommendation, we have included the reference and also added a statement on the wild type variant in the discussion section Line 359.

I also believe that the fact that we have focused on 2020 as a time frame should be considered a limitation, and that the considerations of what has been mentioned cannot be confirmed as a result.

Response: We have now included it as one of the limitations.

Round 2

Reviewer 3 Report

The authors made advances in the new version of the submitted manuscript. In my view, it presents data on the effect of COVID-19 in an article that presents a clear question and results. Perhaps the main limitation of the study is a lack of originality, considering that it does not present new outcomes compared to other articles in the literature. Perhaps in some way the publication can help in dealing with future problems of virus outbreaks.

Described previously.

Author Response

Comments and Suggestions for Authors

The authors made advances in the new version of the submitted manuscript. In my view, it presents data on the effect of COVID-19 in an article that presents a clear question and results. Perhaps the main limitation of the study is a lack of originality, considering that it does not present new outcomes compared to other articles in the literature. Perhaps in some way the publication can help in dealing with future problems of virus outbreaks.

Response: We appreciate your comments. We hope that this publication will help in improving the knowledge in the field of the study.

Reviewer 4 Report

Dear Authors,

I appreciate the amendments; they have undoubtedly improved the quality of the manuscript. While reading this second draft, I noticed several things that need to be evaluated and fixed.

Materials and Methods

Figure 1: It must be expanded in order to be properly consulted.

Lines 181–185: Are you certain that the Pearson Chi-square test was used in the multivariate analysis? Was the logistic regression method used instead for the multivariate analysis? Moreover, I would like you to declare the method you used to select the variables (stepwise, forward, backward, hierarchical, or other); it is essential to specify the method used.

Results

Tables 1, 3, and 4: What is the "Number" column for? I'm still perplexed as to why this column gets repeated.

Tables 5 and 6: I'd like to emphasise the ambiguity of the approach and test utilised once more. Given that probably logistic regression was performed in both circumstances, I do not believe the presence of an asterisk in these two tables is justified. Furthermore, I feel that characteristics like gender and age should always be retained, even if they are not statistically significant.

Discussion

Lines 304–305: Thank you for clarifying what was implausible in terms of dates. The webpage must be included as a reference.

Lines 344–345: I believe what I offered for further examination was misinterpreted. The data evaluated in the study are referred to 2020, so contextualization is required. In particular, when it comes to mortality and intubated subjects, but more broadly, when it comes to subjects for whom SARS-CoV-2 infection has resulted in severe disease outcomes, it is important to remember that your study was conducted during the period when the wild-type was prevalent, but it is also important to remember that each variant is characterised by different profiles in terms of transmissibility and pathogenicity, resulting in different clinical outcomes (evidenced by reference previously suggested). As a result, the findings of your study should be interpreted with caution because they correspond to wild-type dominant period rather than multiple periods. If the study had been conducted during the period of omicron dominance, the most severe outcomes (namely admission in ICU, intubation, and death) would have most likely been lower than those found in your study. What I asked for with the previous comment, is a part based on these and other statements. I would consider inserting it in the part related to limits, because in fact it is a limit.

Author Response

Response to the Comments of Reviewer 4

Comments and Suggestions for Authors

Dear Authors,

I appreciate the amendments; they have undoubtedly improved the quality of the manuscript. While reading this second draft, I noticed several things that need to be evaluated and fixed.

Response: Thank you very much for your valuable observation. We have tried our level best to further improve the manuscript. Hope we have met your expectations.

Materials and Methods

Figure 1: It must be expanded in order to be properly consulted.

Response: We are thankful to the respected reviewer for this comment. Figure 1 is expanded now with the addition of more details on the exclusion reasons at the screening stage.

Lines 181–185: Are you certain that the Pearson Chi-square test was used in the multivariate analysis? Was the logistic regression method used instead for the multivariate analysis? Moreover, I would like you to declare the method you used to select the variables (stepwise, forward, backward, hierarchical, or other); it is essential to specify the method used.

Response: The respected reviewer is very valuable to us. We have made the following corrections in the manuscript based on this comment:

  1. The data analysis section is re-written to clarify the use of the Chi-square test. The following statement is now included.

“Univariate descriptive analysis of the socio-demographic features of the study sample using Pearson Chi-square test was carried out”.

  1. We have used multinomial regression analysis to determine the predictors of the two outcomes ‘intubation and fatality’. The statement in section 2.3 is corrected now as follows:

Multinomial regression analysis was done to calculate the odds ration as a measure of predictors for dependent variables ‘intubation’ and ‘fatality’ in COVID-19 patients.

  1. In the multinomial analysis, we used references within the variables that were given in the original version of the manuscript, however, during revision, it was removed based on recommendations of some of the reviewers (they pointed out that the table is clear even without that information), and it was given as level 1 and 2. Please find the tables that were included before.

Predictors for fatality in COVID-19 ICU admitted patients.

Characteristics

Level 1

Level 2

Odds Ratio

Confidence interval

P value*

Lower

Upper

Sex

Male

Female

3.42

2.38

5.49

0.021

Intubation

Yes

No

1.67

1.21

2.34

0.034

Diabetes and hypertension

Yes

No

7.85

4.31

9.32

0.001

Diabetes

Yes

No

5.43

3.21

7.21

0.001

Hypertension

Yes

No

4.21

2.98

5.32

0.001

Established CV diseases

Yes

No

3.81

2.87

5.98

0.001

Smoking

Yes

No

4.32

2.11

6.21

0.001

Predictors for intubation in COVID-19 ICU admitted patients.

Characteristics

Level 1

Level 2

Odds Ratio

Confidence interval

P value*

Lower

Upper

Sex

Male

Female

2.92

1.88

3.69

0.021

Diabetes and hypertension

Yes

No

6.29

4.98

8.22

0.001

Diabetes

Yes

No

5.64

3.66

6.55

0.001

Hypertension

Yes

No

4.71

2.83

5.72

0.001

Established CV diseases

Yes

No

1.91

1.07

2.76

0.001

Insulin

Yes

No

3.93

2.79

5.77

0.001

Furosemide

Yes

No

3.65

2.81

4.89

0.001

Age more than 50 years

Yes

No

1.81

1.76

3.21

0.034

We believe that tables 5 and 6 in the current form, are indeed clear to explain this aspect, even without the level 1 and level 2 descriptions.

Results

Tables 1, 3, and 4: What is the "Number" column for? I'm still perplexed as to why this column gets repeated.

Response: It was just a serial number; we have now removed it. Thanks for your observation.

Tables 5 and 6: I'd like to emphasise the ambiguity of the approach and test utilised once more. Given that probably logistic regression was performed in both circumstances, I do not believe the presence of an asterisk in these two tables is justified. Furthermore, I feel that characteristics like gender and age should always be retained, even if they are not statistically significant.

Response: We regret this mistake, we have now removed the asterisk from both tables, the response is already given in detail above.

Discussion

Lines 304–305: Thank you for clarifying what was implausible in terms of dates. The webpage must be included as a reference.

Response: Thanks for your recommendation. We have included it as a reference now.

Lines 344–345: I believe what I offered for further examination was misinterpreted. The data evaluated in the study are referred to 2020, so contextualization is required. In particular, when it comes to mortality and intubated subjects, but more broadly, when it comes to subjects for whom SARS-CoV-2 infection has resulted in severe disease outcomes, it is important to remember that your study was conducted during the period when the wild-type was prevalent, but it is also important to remember that each variant is characterised by different profiles in terms of transmissibility and pathogenicity, resulting in different clinical outcomes (evidenced by reference previously suggested). As a result, the findings of your study should be interpreted with caution because they correspond to wild-type dominant period rather than multiple periods. If the study had been conducted during the period of omicron dominance, the most severe outcomes (namely admission in ICU, intubation, and death) would have most likely been lower than those found in your study. What I asked for with the previous comment, is a part based on these and other statements. I would consider inserting it in the part related to limits, because in fact it is a limit.

Response: We regret the misinterpretation. We have included this aspect in the limitation and included a reference as per the suggestion of a respected reviewer. The following phrase is now included in the limitation.

Although the study's main goals were attained, some limitations must be addressed before proceeding with the current findings. Data from 2020 are presented in this analysis; at that time, a COVID-19 wild-type variant was prevalent and was held accountable for the rapid spread of the illness and poor clinical outcomes [55]. Because different COVID-19 variants differed in terms of transmissibility and pathogenicity, our study results would have been different if it had been undertaken during the period of omicron dominance. For that reason, the results of this study should be viewed with caution, as they cannot be generalized to other time periods.
